# Cooperation and Competition Impact Environmental Action: An Experimental Study in Social Dilemmas

**Daniel Curtin and Fanli Jia \*** 

Department of Psychology, Seton Hall University, South Orange, NJ 07079, USA; danielcurtin85@gmail.com
\* Correspondence: fanli.jia@shu.edu; Tel.: +1-(973)-275-2708

**Abstract:** Previous research about social dilemmas has identified cooperation as a possible underlying facilitator of proenvironmental behavior. However, there has been no discussion about how manipulating cooperation and competition could influence environmental action experimentally. The current study filled this gap in previous literature by manipulating cooperation and competition in a group of 155 participants and comparing their respective environmental actions. Participants were randomly placed into one of three conditions and primed by writing a short passage regarding a significant personal experience where they acted cooperatively, competitively, or neutrally. It was found that those in the cooperative priming group scored significantly higher on environmental participatory action than people in the competitive priming group. However, no difference was found on environmental leadership action. The results indicated that participatory environmental actions are relatively easier to change, as the threshold for interest in them is much lower than leadership environmental actions.

**Keywords:** cooperation; competition; environmental action; experiment; social dilemma

## 1. Introduction

The interest surrounding issues of sustainability and the environment is growing in today's psychological research. Recently, Di Fabio [1–3] developed a new framework of the Psychology of Sustainability and Sustainable Development. It provided a transdisciplinary perspective of integrating both sustainability and psychological science in understanding behaviors that promote sustainable leadership and proenvironmental behavior [1–3]. The conclusion being drawn from these psychological studies in sustainability has indicated that, as humans, working collectively within our societal and organizational structures is essential to achieving a heathier planet [1–4]. Thus, it is important to apply the psychological process accounting for personality traits, individual values, emotion, and motivations that related to behavioral and decisional processes in sustainability [1–3].

Researchers in psychology and sustainability science have studied the connection between cooperation and environmental issues using the theory of social dilemmas [5–7]. A social dilemma is a condition in which immediate individual or competitive interests and long-term cooperative interests are in conflict [7]. One common social dilemma is a person's decision to care for or neglect the environment. Past research has explored this scenario and concluded that how a person chooses to act while in a social dilemma can reveal that person's social values. For example, people tend to view environmentally conscious individuals as cooperative, prosocial, and self-transcendent [7–10], because their decisions are beneficial for both the environment and the individual. In contrast, people tend to view less environmentally conscious individuals as competitive and self-interested, since they often exploit resources for monetary gain at the cost of environmental degradation [8,11]. Past studies have validated the relationship between the social values of cooperation and competition and concerns for environmental issues, and many studies have examined personal preference for

cooperation or competition (e.g., social value orientation) in relation to proenvironmental behavior. Researchers have applied correlational design and used either a survey [5,6] or game theory [7,12] to collect data. However, there is limited research on experimental design, and on whether cooperation in recalling personal events causes the intention of proenvironmental behaviors. Thus, the present study examines the causal relationship between a social dilemma (cooperation versus competition) and proenvironmental actions by priming individual autobiographical cooperative/competitive memories.

## 1.1. Cooperation and Competition in Social Dilemmas

Research on cooperation and competition is often completed through models of real-world interdependent issues known as social dilemmas [13]. In the social dilemmas, each actor (e.g., individual, company, or nation) has an incentive to take resources to maximize their own immediate gain, while the delayed cost of doing so will be shared amongst the others until the resource is depleted [14]. An example is open-sea fishing [11], which is relatively available to all, but has a finite supply of resources. As each actor fishes from the sea to maximize their own earnings, fish are depleted until the shared resource system is overfished and destroyed. In this type of scenario, some people will act prosocially and cooperatively to ensure that the action is beneficial for both the environment (e.g., cleaner waterways) and the individual (e.g., cleaner water to drink); in contrast, others will act selfishly and competitively for individual benefit (e.g., exploiting resources for monetary gain), at the cost of environmental degradation (e.g., fish extinctions from overfishing) [11–16]. Social dilemmas are also commonly applied to other real-life environmental issues like water shortages [15] and energy crises [16].

While other researchers [14] have argued that humans are too inherently selfish to manage a shared resource system, many economists [17], psychologists [18], and biologists [19] argue that people are not completely selfish because they often choose to care for the well-being of others. Through several economic models, Monroy et al. [17] explored why people make this choice, and found that only a prosocial agent (e.g., cooperation) is required to avoid the tragedy of the commons, regardless of other agents' behavior [17]. In another study, Kaiser and Burka [20] found that over ninety percent of their proenvironmental participants were cooperative. On the contrary, only nine percent of their proenvironmental participants were selfish [20]. Thus, increasing prosocial values, such as cooperation, could mitigate the amount of short-term selfishness, which could prevent the depletion of common-pool resources [17] and encourage environmental actions [20].

## 1.2. Environmental Action

Environmental action is often used interchangeably with the terms proenvironmental behaviors and environmental activism [21]. A proenvironmental activist is someone who demonstrates a broad range of proenvironmental behaviors, as well as holds environmental consciousness (value and belief) [22]. Recently, Alisat and Riemer [21] defined environmental action as "intentional and conscious civic behavior that is focused on the systematic causes of environmental problems and the promotion of sustainability through collective efforts." In their approach, environmental action is collective and cooperative. They developed a scale that measures levels of engagement in environmental actions, extending from low-level civic action to highly engaged political or organizational action. Less involved civic actions could consist of something as simple as searching for information on the internet regarding environmental issues, whereas a more highly involved action could be organizing a protest to bring awareness to environmental issues [21]. Alisat and Riemer [21] separate everyday environmental participatory action from environmental leadership behaviors. Perhaps the understanding that cooperation will yield a common good underlies many instances of proenvironmental action [23]. Similarly, Dono et al. [22] make a distinction between environmental activism and everyday environmental practices based upon the collective nature of environmental action. Accordingly, this study adopted a division between environmental action and daily personal practices. This division is empirically based in research that suggests that personal attempts to alter

daily behaviors to help the environment (e.g., buying a more energy-efficient vehicle) have been relatively unsuccessful [23], and in research that argues that meaningful environmental action is based in collective behaviors.

### 1.3. Relationship between Social Dilemmas and Environmental Action

Past research has demonstrated the importance of investigating social values and environmental actions [5–12]. Joireman describes the long cycle of competitiveness that has led to our current global environmental status [7]. He points out that it has taken countless environmentally selfish acts by countless individuals to get to our current level of global degradation [7]. These acts can be viewed as decisions between immediate selfish and long-term cooperative interests. In many environmental decisions, a person must choose between competitive or cooperative interests, and immediate or long-term interests. A situation in which immediate selfish competitive interests and long-term cooperative interests are at odds is considered a social dilemma. Therefore, examining environmental issues through the lens of a social dilemma could illuminate the relationship between cooperation and environmentalism [24,25].

Kramer, McClintock, and Messick [26] suggest that individuals differ in the way they approach environmental social dilemmas because of two main factors: social values and the structural characteristics of the specific dilemma. A social value refers to an individual's preference for cooperation or noncooperation. Structural characteristics are defined by how mildly or severely the social dilemmas are modeled. For example, a social dilemma could be relatively mild (e.g., running out of art supplies at a school) or quite severe (e.g., starvation). Kramer and his colleagues explored these structural characteristics using a decomposed game procedure, which is a laboratory game experiment designed to model real-life social dilemmas. Based upon participants' social values determined in the procedure, Kramer and his colleagues separated participants into two categories: cooperators and noncooperators [26]. They then compared how the two groups (cooperators and noncooperators) acted in a resource conservation task designed to measure proenvironmental behavior [26]. A resource conservation task was set up so that participants would decide how many valuable points (i.e., fish) they would take from a collective resource pool (i.e., the ocean). The amount of real-life money a participant would earn in the study was based upon the number of points they took from the collective pool. Much like real-life environmental social dilemmas, as the participants took more points for themselves, the collective resource pool (i.e., ocean) was depleted until no points (i.e., fish) were left. They found a significant main effect for social value, with cooperators taking fewer resources for themselves. Specifically, Kramer, McClintock, and Messick argue that cooperation in social dilemmas could be linked to personal restraint in environmental decision-making, based on their results [26].

In addition, Joireman, Van Lange, Kuhlman, Van Vugt, and Shelley [27] investigated the relationship between social value orientation (labeling participants as cooperative versus competitive) and environmental decision-making by giving participants a set of surveys. They sampled commuters in both Dutch train stations and gas stations. Participants at the two locations were given identical surveys regarding their commuting preferences and social value orientations. Joireman et al. [27] found that those with "other-oriented" (cooperative) concerns were increasingly likely to have the desire to use public transportation when compared to those with "self-oriented" (competitive) concerns.

Similarly, Kaiser and Burka [20] found that environmentalists (participants scoring highly in environmental action) acted more prosocially than their nonenvironmental participants. They compared participants previously labeled as prosocial to those that they labeled as proselfish. A large majority of their prosocial participants were labeled as highly environmentally engaged (90.2%), while only 9.8% of their proselfish participants were labeled as highly environmentally engaged. Additionally, they found that their proselfish participants scored lower in environmental engagement than their prosocial participants.

In a recent mixed-methods study, Jia et al. [8] examined how different moral values relate to an individual's tendency towards environmental involvement. Three types of moral values were

found: self-transcendence, self-interest, and mixed values. Researchers found that participants who endorsed self-transcendent moral values (e.g., universalism, concern, and caring) scored significantly higher on environmental involvement than the other two identities. In contrast, people who endorsed self-interested moral values (e.g., achievement and self-direction) scored the lowest on environmental involvement (also see Author's work [28] to access the method and the data).

### 1.4. Present Study and Hypotheses

A major shortcoming of the current literature investigating the relationship between social dilemmas and environmentalism is that many researchers have employed a correlational or quasi-experimental design. For instance, Dono et al. [22] used a factor analysis to illuminate a predictive, correlational relationship between environmental activism and aspects of social identity [22]; Jia et al. [8–10] explored the relationship between prosocial values and environmental engagement using cluster analyses and narrative examples; Joireman et al. [27], as well as Kaiser and Burka [20], labeled the preferences for cooperation and competition. Unfortunately, because of their correlational design, those studies were unable to suggest the possible causes or nature of this relationship. Only one study has explored a causal relationship using priming in the literature of environmental psychology. Priming is an experimental technique where participants are exposed to cues (i.e., words, objects, memories, etc.) to later trigger unconscious memories or actions. In fact, Zaval, Markowitz, and Weber [25] successfully primed participants to have an increased propensity to donate to environmentally related issues. They asked participants to write a short essay describing what the participants wish to be remembered for, an exercise designed to prime participants to have stronger legacy motives. However, no study has explored a causal relationship between cooperation and environmental actions. Additional research must be completed on the experimental typology so that conclusions can be drawn about the true nature and direction of any relationship present between cooperation and environmentally relevant behaviors. This current study utilized a similar priming mechanism by asking participants to write about their autobiographical memories to facilitate cooperation and competition.

This study addressed the gap in the previous literature by conducting an experiment priming social dilemmas. Participants were primed by writing narrative stories. This priming method has been used in recent environmental psychological studies [25]. It was expected that, if participants were primed by writing cooperative stories, they would score higher in environmental action. Participants primed by writing competitive stories were expected to score lower on environmental action. In addition, environmental action was considered a multifaceted construct including daily participatory practices (e.g., recycling) and leadership actions (e.g., protesting). Participatory actions are easy to adopt at an early stage, whereas leadership actions are more difficult to adopt as they require a high level of determination. Group differences on priming were explored in each construct of environmental action. We expected that participatory proenvironmental actions would be easier to prime than leadership actions.

## 2. Method

### 2.1. Participants

The participants were recruited from a Psychology Research Participation System at Seton Hall university, United States. Sign-up was voluntary, and participants earned partial credit towards class research participation requirements. To achieve an effective sample size, a power analysis was conducted using G power, which determined that 0.95 power could be reached with 129 participants (three groups, two measures, repeated measure, and between-factor MANOVA). In 2018, 155 participants were recruited from online research participation sign-up software. The participants included 115 women and 39 men, aged 18 to 51 years old (M = 20.53, SD = 2.82). The cooperative priming group consisted of 36 women and 14 men (N = 50), the competitive priming group consisted of 41 women, 13

men, and one transgender person (N = 55), and the neutral priming group consisted of 38 women and 12 men (N = 50). All participants were actively enrolled at the university at the time of the study. The study was approved by the Institutional Review Board at the university.

*2.2. Procedures and Measures*

The study is a between-subject design with two experimental groups and one control group. In the experimental groups, participants were asked to recall and write about past autobiographical memories when they were cooperative (cooperative priming condition) or competitive (competitive priming condition). The control group was asked to recall an unrelated memory.

The study was administered to a small group through computer-based software (Qualtrics). Using the "Randomizer-Branch If" function in Qualtrics, researchers created three random blocks (the cooperative, competitive, and neutral conditions) to ensure that participants had an equal chance of being assigned to any of the given conditions. After completing the priming experiment, the participants were subsequently evaluated with a set of questionnaires.

To prime our participants, we asked them to write a brief passage about a past personal experience in which they were cooperative (cooperative priming condition) or competitive (competitive priming condition). Participants assigned to each group responded to the corresponding prompts (see below). After the priming procedure, subjects took a survey to assess environmental action. Additionally, prior to administering our questionnaire, we conducted a manipulation check to ensure that the priming procedure was successful.

### 2.2.1. Cooperative Priming Condition

"I'd like you to recall a stand-out event in your life when you cooperated with others. This would be a time when you sacrificed your own gain (i.e., money, happiness, prestige) for the collective gain of a group (i.e., family, coworkers, friend group, society). It should be an important moment or episode in your own life story in which you experienced positive feelings (i.e., joy, excitement, peace, happiness) cooperating with others. Choose one event or episode that is fundamental to your life. Please exclude sports related events.

Describe it in detail–making sure to include what led up to the event so that it can be understood in context. Also include when and where it happened, who was involved, what you were thinking and feeling during the event, why it is important to you, and what impact the event has had on your life."

### 2.2.2. Competitive Priming Condition

"I'd like you to recall a stand-out event in your life when you competed against others. This would be a time when you maximized your own gain (i.e., money, happiness, prestige) against the relative gain of a group (i.e., family, coworkers, friend group, society). It should be an important moment or episode in your own life story in which you experienced positive feelings (i.e., joy, excitement, peace, happiness) competing against others. Choose one event or episode that is fundamental to your life. Please exclude sports related events.

Describe it in detail–making sure to include what led up to the event so that it can be understood in context. Also include when and where it happened, who was involved, what you were thinking and feeling during the event, why it is important to you, and what impact the event has had on your life."

### 2.2.3. Control Group

"I'd like you to recall a stand-out event in your life when you interacted with others. It should be an important moment or episode in your own life story in which you experienced positive feelings (i.e., joy, excitement, peace, happiness) with others. Choose one event or episode that is fundamental to your life. Please exclude sports related events.

Describe it in detail–making sure to include what led up to the event so that it can be understood in context. Also include when and where it happened, who was involved, what you were thinking and feeling during the event, why it is important to you, and what impact the event has had on your life"

Participants were asked to exclude sports-related events because team sports create a rare situation where cooperation can be competitive, and competition can be cooperative. Recalling a significant past personal life experience required participants to access their autobiographic memory. This process promotes the activation of specific memories in the autobiographical memory system, and primes other related memories [29]. Specifically, autobiographic memory could facilitate associations wherein episodes are associated by the same or similar content [30]. This method has been applied to previous research in environmental psychology [25]. Our design paralleled this concept; it was expected that recalling a significant cooperative or competitive life event could cause associations of cooperative or competitive thoughts.

### 2.2.4. Manipulation Check

To ensure that the priming procedure was effective, we used a manipulation check to evaluate cooperative and competitive values. A brief questionnaire of cooperative and competitive values [31] was administered to participants after priming. It included three items that measured cooperativeness and three items that measured competitiveness. Participants responded to the questionnaire on a Likert-type scale from 1 (strongly disagree) to 7 (strongly agree). A sample item of cooperativeness is, "In order to succeed at work, a person must cooperate with their partners." A sample item of competitiveness is, "Even in a group working towards a common goal, I still want to outperform others." We expected participants primed in the cooperative condition to score higher on cooperativeness. We expected participants primed in the competitive condition to score lower on cooperativeness. Additionally, we expected the participants in the control group to score moderately on cooperativeness.

### 2.2.5. Environmental Action

To measure environmental action, participants completed an environmental action scale (EAS) questionnaire [21]. The environmental action scale consists of 18 items that assess a range of environmental actions. Nine items assess daily participatory actions (e.g., educating oneself about environmental issues); the other nine items assess organizational leadership action (e.g., organizing a proenvironmental protest). Participants responded to questions in the following format: "In the next six months, how often, if at all, do you plan to engage in the following environmental activities and actions?" Items were rated on a 5-point scale ranging from one (never) through to five (frequently). The Cronbach's alpha for this scale was 0.88 for the current study.

This experiment is part of the first author's unpublished Masters Thesis project, which also measured environmental identity [32] and environmental attitude (New Ecological Paradigm) [33]. In the original study, researchers immediately presented participants with the environmental action scale once they completed the primer (writing autobiographical memories about cooperation or competition). We did not expect a priming manipulation to affect the other environmental constructs, because the priming effect is very brief. We ran separate analyses to check if there was a priming effect. The results confirmed our expectations. There was no significant difference across groups on environmental attitude, $F_{(2,155)} = 0.006$, $p = 0.995$, partial $n^2 = 0.000$; or on environmental identity $F_{(2, 155)} = 0.433$, $p = 0.649$, partial $n^2 = 0.006$. Thus, the original study differs from the current one, because it focused on the priming effect of environmental action.

## 3. Results

### *3.1. Descriptive Analyses and Manipulation Check*

After being exposed to the manipulation, the participants who wrote cooperative stories reported higher cooperation ($M = 6.06$, $SD = 0.68$), compared with those in the competitive condition ($M = 5.71$,

$SD$ = 0.91), $p$ = 0.018. However, participants in the control group ($M$ = 5.96, $SD$ = 0.64) did not significantly differ from either priming condition. On average, participants wrote 191.03 words in each story. The cooperative group responded to the prime with two main themes; sacrificing for family and volunteering. The theme of sacrificing for family was centered around participants writing about personal sacrifices they made for the benefit of their families. The theme of volunteering consisted of participants writing about times that they had given up happiness, money, time, and so forth to volunteer for causes. In the cooperative priming group, 16 participants out of 50 wrote about sacrificing for family, 24 participants out of 50 wrote about volunteering, and the remaining 10 wrote about miscellaneous topics such as playing in a marching band. The competitive priming group responded to the prime with two main themes; work and school. The work theme consisted of situations in which participants competed to further their careers, while the school theme consisted of situations in which participants competed to succeed academically. In the competitive priming group, 19 participants out of 55 wrote about work, 24 participants out of 55 wrote about school, and the remaining 12 wrote about miscellaneous topics such as competing in spelling bees. The control priming group responded to the prime with two main themes; stories about vacations or trips, and school. The vacation or trip theme consisted of times in which participants had gone on important vacations with friends or family that had an impact on their lives. The school theme consisted of important academic experiences. Interestingly, many of the school-related responses were centered around graduation. In the control priming group, 27 participants out of 50 wrote about vacations or trips, 11 out of 50 participants wrote about school, and the remaining 12 wrote about miscellaneous topics such as jobs.

### 3.2. Main Results

An initial MANOVA was run with three groups (cooperative priming condition, competitive priming condition, and neutral priming condition) and the dependent variable (environmental action). Means and standard deviations are reported in Table 1. There was no significant difference among groups in environmental action, $F(2,155)$ = 1.476, $p$ = 0.232, partial $n^2$ = 0.019. However, a post hoc Least Significant Difference (LSD) test on the priming groups revealed that people in the cooperative priming group scored marginally higher on environmental action than people in the competitive priming group ($p$ = 0.049). There were no statistically significant differences between the cooperative priming group and the control group ($p$ = 0.137) or between the competitive priming group and the control group ($p$ = 0.649).

**Table 1.** Means and standard deviations of the environmental action subscales.

|  | Participatory Action | Leadership Action | Env. Action |
|---|---|---|---|
| Cooperation Priming | 2.84 (0.67) | 1.46 (0.53) | 2.23 (0.57) |
| Competition Priming | 2.47 (0.67) | 1.40 (0.65) | 1.99 (0.59) |
| Control | 2.56 (0.89) | 1.42 (0.51) | 2.05 (0.66) |

In order to further analyze environmental action, the environmental action scale was split into two subscales, following previous literature that has shown that environmental action can be categorized into daily participatory actions and leadership actions [21]. A one-way ANOVA was run to test group differences (cooperative priming condition, competitive priming condition, and neutral priming condition) in the two environmental action subscales (participatory environmental actions and leadership environmental actions). The results indicated a significant difference by priming groups on participatory environmental action $F(2,155)$ = 3.474, $p$ = 0.033, partial $n^2$ = 0.044, but no statistical difference by priming condition on leadership environmental action $F(2.155)$ = 0.199, $p$ = 0.819, partial $n^2$ = 0.003. A post hoc independent $t$ test indicated that people in the cooperative priming group ($M_{coop}$ = 2.84, $SD$ = 0.67) scored significantly higher ($t$ = 2.83, $p$ = 0.006, Cohen's $d$ = 0.55) in participatory environmental action than the competitive priming group ($M_{comp}$ = 2.47, $SD$ = 0.67). However, no

statistically significant differences emerged between the cooperative priming group and the control group ($p = 0.057$) or the competitive priming group and the control group ($p = 0.559$).

## 4. Discussion

Environmental issues are often modeled in psychological research with social dilemmas, with the tragedy of the commons resource being modeled most often [13]. The social dilemma involves a conflict between individual (or competitive) interests and group (or cooperative) interests. Similarly, many environmental issues (e.g., water usage or overfishing) can be framed as a conflict between competitive (individual) and cooperative (group) interests. Thus, manipulating cooperation and competition was expected to influence scoring on environmental measures in the present study. The cooperative manipulation group was hypothesized to score higher on environmental action than the competitive group. However, the hypotheses were only partially supported: the cooperative priming group scored marginally higher on environmental action than people in the competitive priming group. Furthermore, participants in the cooperative priming group scored significantly higher on participatory environmental action than the competitive priming group. No group difference was found in environmental leadership action.

The group differences in environmental action that emerged between the cooperative and competitive priming groups are consistent with past literature. A previous study utilized a similar priming method to enhance environmental action [25]. Zaval et al. found that primed participants donated to environmental causes at a significantly higher rate than unprimed participants. Interestingly, group differences reported by similar priming studies also exhibited borderline significance [34]. This may reflect the nature of priming for social values. Another past study conducted by Kenis and Mathijs also argued that meaningful environmental actions are based in collective behaviors [23]. They found a positive correlation between responses to environmental action questionnaires and prosocial values [23].

The responses to environmental action were investigated further by splitting the environmental action measure into two subscales: participatory actions and leadership actions [21]. Participatory actions are categorized as involvement in environmental actions through established methods, while leadership actions include taking organizational roles or managing environmental initiatives. An example of a participatory environmental action is reading a monthly newsletter to inform oneself about current issues, while an example of a leadership action is organizing a recycling drive. These two subscales were identified by Alisat et al. during the development of their overall environmental action scale [21]. They theorized that participatory actions are the first type of environmental action that a person adopts, while leadership actions are developed over time as personal interest in environmental issues grows. The results indicated a significant difference by priming groups on participatory environmental action. Participants in the cooperative priming group scored significantly higher on participatory environmental action than the competitive priming group. These results are supported by Alisat et al. [21]. They suggest that participatory actions are easier to adopt at an early, less-engaged stage of environmental action [21]. If the priming mechanism was indeed not particularly strong, the results would be expected as such. Participatory environmental actions are relatively easier to prime, as the threshold for interest in them is much lower than for leadership environmental actions. As expected, there was no statistical difference by priming condition on environmental leadership action. Environmental leadership actions take a high level of involvement and time investment, and it would likely require a more robust prime to impact a person's behavior in this way.

This study had some limitations. First, it is possible that participants did not take the priming task seriously, or that they did not follow the prompt closely enough to become engrossed in their past autobiographic memories. One major issue is that the priming manipulation in the present study may not enhance the cooperative score. It was found that people in the cooperative priming condition did not score higher in cooperative values than those in the control condition. The findings of the study should be interpreted with caution. It may be that the manipulation check by Lu et al. [31] assessed

cooperation as a personality trait. In the current study, we considered cooperation as a state that can be primed. However, we were unable to test this assumption. Future research should investigate the cooperation/competition social dilemma using a more robust priming mechanism. One possible solution is to add a vital control variable (e.g., social value orientation) to infer the consequence of either priming or individual difference. Another possible solution is to design a priming method to distinguish cooperation as either a trait or a state. This may trigger a theoretical discussion about whether or not the priming of cooperation has a strong enough effect to permanently change people to become proenvironmental.

A second limitation of the priming task in this study is that it was relatively brief and open-ended. Previous research suggests that priming is improved by both repetition of a priming stimulus and the duration of the prime [35]. As a result, the limitation of our priming mechanism could be abetted with a longer prime or repetitive prime. Participants could write a series of short stories or a single longer story. In addition, the contents of the prime could be improved with a more specific priming instruction. Much of the content of the stories did not follow the exact operational definitions of cooperativeness or competitiveness. Some content did not involve sacrifice for the collective gain of a group or a conflict with the collective gain of the group. However, it is important to maintain enough flexibility that participants are able to think of stories from their lives that match our definitions. If the instructions become too specific and rigid, many participants may be unable to think of real stories from their lives.

Additionally, the nonsignificant results could be explained by demand characteristics. Participants may have been responding to the questionnaires in a manner that would make them seem more cooperative, since cooperation is seen as more socially desirable than competition by many American college students [36]. An open-ended question at the end of the questionnaire asked participants what they perceived the purpose of the study to be, and 18 out of 50 participants in the cooperative group, 22 out of 55 participants in the competitive group, and 9 out of 50 participants in the control group indicated that they knew the study was looking into the effect of cooperation and competition on environmental values. An example of a response to the open-ended question from a participant in the competitive condition is:

"The purpose of this study is to show the relationship between morals and caring for environmental issues. I believe that perhaps if one enjoys competition and is more of a selfish person the assumption is that they would not care for environmental issues and vice versa."

The response does not indicate any knowledge of the priming mechanism. However, it does indicate a basic belief that competitive values are linked to carelessness towards the environment. This belief may have impacted the way participants responded to the questionnaire.

One possible future direction to improve upon the study would be to emphasize the interdependency of the social dilemma during the priming mechanism. Interdependent situations are where two or more people mutually rely upon each other for an outcome. Dawes notes that cooperation increases when people understand that their actions affect other people and that other people's actions affect them [13]. A future direction could be to make interdependence in social dilemmas more salient. For example, a confederate could be used to make the social dilemma feel more genuine.

Another future consideration to improve upon this study would be to diversify the sample. The current study's sample involved 155 college students from a private university in New Jersey. There is a growing concern in science that samples like this, which are mainly composed of participants who are WEIRD—Western, educated, industrialized, rich, and democratic—are overused [37]. As a result, the generalizability of the study's results needs to be interpreted with caution. Future research must therefore seek to understand environmental behaviors in non-WEIRD populations, and examine how social and cultural factors moderate the effectiveness of existing environmental behaviors [38–42]. One possible solution is to utilize the national/international data sets on proenvironmental action, which are available based on an open science policy [43–50]. This research could be used to identify

cultural factors (e.g., urban/rural, socioeconomic status ethnicity, and religion) that may moderate the relationship between values (e.g., cooperation) and environmental action [51,52].

Although priming cooperation does not influence general intentions to act proenvironmentally, an important contribution of the current study is the disentanglement of the two types of proenvironmental actions: participatory actions and leadership actions. Participatory actions are easy to adopt at an early stage, whereas leadership actions are more difficult to adopt because they involve a high level of determination. The different levels of adopted actions or intentions were illustrated in the Theory of Planned Behavior (TPB) [53], and were supported by research into waste management [54]. The TPB argued that the effects of an individual's perceived ability (easy or difficult) to adopt a behavior predicted intentions to act proenvironmentally or not. Similarly, past studies about waste management have found that perception of the task as easy or difficult was the main predictor of recycling behaviors [54]. Therefore, future research in the field of environmental psychology should also distinguish between participatory actions and leadership proenvironmental actions, to ensure that different levels of adopted actions are considered. Ideally, researchers can preregister hypotheses stating that cooperation versus competitiveness will specifically influence low-cost environmental participatory actions rather than leadership actions.

In the current study, we examined proenvironmental actions as acting with intention or as a response to a concern. However, Steg and Vlek suggest that environmental concerns are not strongly associated with actual proenvironmental actions [55,56]. As a result, future research should examine the specific behavioral component of proenvironmental participatory actions. For example, researchers could create an environmental website and ask participants to indicate how much money they would like to donate to the environmental organization, to offer volunteer time for that environmental group, or to provide contact details to be informed about an environmental issue.

The implications of the results could affect how governments or private businesses approach environmental conservation efforts. When recruiting people to support environmental causes, lower-level participatory actions should be easier to elicit than higher-level leadership actions. Environmental initiatives could promote cooperation to achieve greater involvement in participatory environmental actions. Marketing for recycling and other lower-level participatory involvements should be a focus point. Following the theory of Alisat et al. [21], as people become more involved in participatory environmental actions, they could begin to take on environmental leadership roles over time. Additionally, modifying educational programs to include a unit focusing on cooperation in the context of participatory environmental actions may serve as a step forward in mitigating our current environmental issues.

In sum, the results of this study represent an initial step towards addressing the issue of environmentally unfriendly behavior at the individual level. Focusing on individual social values like cooperation could help mitigate climate issues when combined with higher-level efforts in the corporate world.

**Author Contributions:** Conceptualization, D.C. and F.J.; Writing original draft preparation, D.C. and F.J.; Revision, F.J.; Supervision, F.J. All authors have read and agreed to the published version of the manuscript.

**Funding:** The project did not receive any financial support.

**Acknowledgments:** We are grateful to the students who participated in this study. Many thanks go to Dr. Kelly Goedert and Dr. Susan Teague for their comments in designing the study.

**Conflicts of Interest:** The authors declare no conflict of interest.

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
