# Peer review of "Cooperation and Competition Impact Environmental Action: An Experimental Study in Social Dilemmas"

_sustainability, doi:10.3390/su12031249_

Round 1
Reviewer 1 Report
I think the actual design of the participant sampling for the research objective is not adequate. The identification of the participants representing their inclined social value should not be randomly selected. The research question is about finding an individual's preference for cooperation or competition expressed in behavior and leadership - the participants should therefore be assigned to one of the three conditions that match with their own preference and condition.
Author Response
Point 1: I think the actual design of the participant sampling for the research objective is not adequate. The identification of the participants representing their inclined social value should not be randomly selected. The research question is about finding an individual's preference for cooperation or competition expressed in behavior and leadership - the participants should therefore be assigned to one of the three conditions that match with their own preference and condition.
Response:
Sorry for not being clear about the research question. The research question is NOT about finding a preference for cooperation or competition expressed in pro-environmental actions. Past studies have examined the preference via either questionnaires or via game theory. In that approach, researchers can identify participants’ preferences for cooperation (prosocial) or competition (proself) and look for relationships (correlational design) between preferences and behaviors.
However, the purpose of the current study is to see if priming (or eliciting) either cooperation or competition affects pro-environmental actions (e.g., participatory actions). The experimental design (random assignment) was applied. Thus, participants have an equal chance of being assigned to any of the experimental conditions. It also helps ensure that any potential differences among the participants are spread out evenly across all experimental conditions.
In the revision, we explicitly stated the research question on page 2.
“many studies have examined personal preference for cooperation or competition (e.g., social value orientation) in relation to pro-environmental behavior. Researchers have applied correlational design and used either a survey [5, 6] or game theory [7, 12] to collect data. However, there is limited research on experimental design, and on whether cooperation in recalling personal events causes the intention of pro-environmental behaviors. Thus, the present study examines the causal relationship between a social dilemma (cooperation vs. competition) and pro-environmental actions by priming individual autobiographical cooperative/competitive memories.”
We also discussed additional studies about the preference and priming in sections 1.3 and 1.4.
“In addition, Joireman, Van Lange, Kuhlman, Vugt, and Shelley [27] investigated the relationship between social value orientation (labeling participants as cooperative vs. competitive) and environmental decision making by giving participants a set of surveys. They sampled commuters in both Dutch train stations and gas stations. Participants at the two locations were given identical surveys regarding their commuting preferences and social value orientations. Joireman et al. [27] found that those with “other-oriented” (cooperative) concerns were increasingly likely to have the desire to use public transportation when compared to those with “self-oriented” (competitive) concerns.”
“Similarly, Kaiser and Burka [20] found that environmentalists (participants scoring highly in environmental action) acted more prosocially than their non-environmental participants. They compared participants previously labeled as prosocial to those that they labeled as proselfish. A large majority of their prosocial participants were labeled as highly environmentally engaged (90.2%), while only 9.8% of their proselfish participants were labeled as highly environmentally engaged. Additionally, they found that their proselfish participants scored lower in environmental engagement than their prosocial participants.”
“In a recent mixed methods study, Jia et al. [8] examined how different moral values relate to an individual’s tendency towards environmental involvement. Three types of moral values were found: self-transcendence, self-interest, and mixed values. Researchers found that participants who endorsed self-transcendent moral values (e.g., universalism, concern, and caring) scored significantly higher on environmental involvement than the other two identities. In contrast, people who endorsed self-interested moral values (e.g., achievement and self-direction) scored the lowest on environmental involvement (also see Author’s work [28] to access the method and the data).”
“A major shortcoming of the current literature investigating the relationship between social dilemmas and environmentalism is that many researchers have employed a correlational or quasi-experimental design. For instance, Dono et al. [22] used a factor analysis to illuminate a predictive, correlational relationship between environmental activism and aspects of social identity [22]; Jia et al. [8-10] explored the relationship between prosocial values and environmental engagement using cluster analyses and narrative examples; Joireman et al. [27], as well as Kaiser and Burka [20], labeled the preferences for cooperation and competition. Unfortunately, because of their correlational design, those studies were unable to suggest the possible causes or nature of this relationship. Only one study has explored a causal relationship using priming in the literature of environmental psychology. Priming is an experimental technique where participants are exposed to cues (i.e. words, objects, memories, etc.) to later trigger unconscious memories or actions. In fact, Zaval, Markowitz, and Weber [25] successfully primed participants to have an increased propensity to donate to environmentally related issues. They asked participants to write a short essay describing what the participants wish to be remembered for, an exercise designed to prime participants to have stronger legacy motives. However, no study has explored a causal relationship between cooperation and environmental actions. Additional research must be completed on the experimental typology so that conclusions can be drawn about the true nature and direction of any relationship present between cooperation and environmentally relevant behaviors. This current study utilized a similar priming mechanism by asking participants to write about their autobiographical memories to facilitate cooperation and competition.”
Reviewer 2 Report
First, I would like to thank the authors for considering my own and the other reviewers' suggestions. While I appreciate the changes the authors have made, I think that some of the issues I mentioned in my previous review persist. Below, I will make some suggestions about how to address them.
I think the manuscript overall would benefit from some proof-reading and editing to improve clarity. This is more of an issue in the introduction and discussion, whereas I found the methods and results sections to be clearer. For example, in the introduction, there is much repetition of the definition of social dilemmas. Instead, the authors could focus more on how priming or framing affect pro-environmental behaviors, or why cooperative versus competitive social values might relate to environmental action. Although I understand that the data reported in this study was collected as part of a larger project that included more measures (which is fine), it's still unclear to me why the authors decided to exclude these measures from the current study. Do they mean that they did not expect a brief priming manipulation to affect the other environmental constructs (i.e., environmental identity and NEP)? This point could also use some clarification. I appreciate that the authors transparently report the comparisons between their experimental conditions and the control condition on the manipulation check. But, the fact that there were no differences between experimental conditions and the control condition on the manipulation check puts into question the interpretation of findings. That is, if people in the cooperative priming condition did not score higher in cooperative values than those in the control condition, what does that tell us about any effects of the priming manipulation? Finally, I think that the authors' argument about cooperative priming influencing participatory, but not leadership, actions makes sense, post-hoc. What would strengthen confidence in this finding would be to replicate their study, ideally pre-registering the hypotheses that cooperation versus competitiveness will specifically influence low-cost environmental actions rather than leadership actions.Author Response
First, I would like to thank the authors for considering my own and the other reviewers' suggestions. While I appreciate the changes the authors have made, I think that some of the issues I mentioned in my previous review persist. Below, I will make some suggestions about how to address them.
Point 1: I think the manuscript overall would benefit from some proof-reading and editing to improve clarity. This is more of an issue in the introduction and discussion, whereas I found the methods and results sections to be clearer. For example, in the introduction, there is much repetition of the definition of social dilemmas. Instead, the authors could focus more on how priming or framing affect pro-environmental behaviors, or why cooperative versus competitive social values might relate to environmental action.
Response #1:
The revised manuscript has been edited by MDPI language editing service. Please see the attached certificate.
Regarding the introduction, thank you for providing additional guidance. We implemented your suggestion in the following sections:
In the revised section 1.1, we cut the repetition of the definition of social dilemmas (the definition was stated in the general introduction). We briefly introduced the social dilemmas and provided an example of “open-see fishing” to demonstrate how competition (proself) and cooperation (prosocial) operated in the social dilemmas.“In the social dilemmas, each actor (e.g., individual, company, or nation) has an incentive to take resources to maximize their own immediate gain, while the delayed cost of doing so will be shared amongst the others until the resource is depleted [15]. An example is open-sea fishing [11], which is relatively available to all, but has a finite supply of resources. As each actor fishes from the sea to maximize their own earnings, fish are depleted until the shared resource system is overfished and destroyed. In this type of scenario, some people will act prosocially and cooperatively to ensure that the action is beneficial for both the environment (e.g., cleaner waterways) and the individual (e.g., cleaner water to drink); in contrast, others will act selfishly and competitively for individual benefit (e.g., exploiting resources for monetary gain), at the cost of environmental degradation (e.g., fish extinctions from overfishing) [11-16]. Social dilemmas are also commonly applied to other real-life environmental issues like water shortages [14] and energy crises [16].”
We expanded the section 1.3 by adding additional studies on why and how social values (cooperative vs. competitive) relates to environmental action.“In addition, Joireman, Van Lange, Kuhlman, Vugt, and Shelley [27] investigated the relationship between social value orientation (labeling participants as cooperative vs. competitive) and environmental decision making by giving participants a set of surveys. They sampled commuters in both Dutch train stations and gas stations. Participants at the two locations were given identical surveys regarding their commuting preferences and social value orientations. Joireman et al. [27] found that those with “other-oriented” (cooperative) concerns were increasingly likely to have the desire to use public transportation when compared to those with “self-oriented” (competitive) concerns.”
“Similarly, Kaiser and Burka [20] found that environmentalists (participants scoring highly in environmental action) acted more prosocially than their non-environmental participants. They compared participants previously labeled as prosocial to those that they labeled as proselfish. A large majority of their prosocial participants were labeled as highly environmentally engaged (90.2%), while only 9.8% of their proselfish participants were labeled as highly environmentally engaged. Additionally, they found that their proselfish participants scored lower in environmental engagement than their prosocial participants.”
“In a recent mixed methods study, Jia et al. [8] examined how different moral values relate to an individual’s tendency towards environmental involvement. Three types of moral values were found: self-transcendence, self-interest, and mixed values. Researchers found that participants who endorsed self-transcendent moral values (e.g., universalism, concern, and caring) scored significantly higher on environmental involvement than the other two identities. In contrast, people who endorsed self-interested moral values (e.g., achievement and self-direction) scored the lowest on environmental involvement (also see Author’s work [28] to access the method and the data).”
In section 1.4, we stated the major shortcoming of the past studies and discussed how priming affects pro-environmental behaviors.“A major shortcoming of the current literature investigating the relationship between social dilemmas and environmentalism is that many researchers have employed a correlational or quasi-experimental design. For instance, Dono et al. [22] used a factor analysis to illuminate a predictive, correlational relationship between environmental activism and aspects of social identity [22]; Jia et al. [8-10] explored the relationship between prosocial values and environmental engagement using cluster analyses and narrative examples; Joireman et al. [27], as well as Kaiser and Burka [20], labeled the preferences for cooperation and competition. Unfortunately, because of their correlational design, those studies were unable to suggest the possible causes or nature of this relationship. Only one study has explored a causal relationship using priming in the literature of environmental psychology. Priming is an experimental technique where participants are exposed to cues (i.e. words, objects, memories, etc.) to later trigger unconscious memories or actions. In fact, Zaval, Markowitz, and Weber [25] successfully primed participants to have an increased propensity to donate to environmentally related issues. They asked participants to write a short essay describing what the participants wish to be remembered for, an exercise designed to prime participants to have stronger legacy motives. However, no study has explored a causal relationship between cooperation and environmental actions. Additional research must be completed on the experimental typology so that conclusions can be drawn about the true nature and direction of any relationship present between cooperation and environmentally relevant behaviors. This current study utilized a similar priming mechanism by asking participants to write about their autobiographical memories to facilitate cooperation and competition.”
Point 2: Although I understand that the data reported in this study was collected as part of a larger project that included more measures (which is fine), it's still unclear to me why the authors decided to exclude these measures from the current study. Do they mean that they did not expect a brief priming manipulation to affect the other environmental constructs (i.e., environmental identity and NEP)? This point could also use some clarification.
Response #2: The reviewer is right. We did not expect the priming effects on later environmental questionnaires (NEP and Environmental Identity) because the priming effect was very brief. We run the analyses to check if there was a priming effect. The results confirmed our expectations. There was no significant difference based on the priming groups: environmental attitude, F(2,155) = .006, p = .995, partial n2 = .000; environmental identity F(2, 155) = .433, p = .649, partial n2 = .006.
We added this clarification on page 10.
“We did not expect a priming manipulation to affect the other environmental constructs because the priming effect is very brief. We ran separate analyses to check if there was a priming effect. The results confirmed our expectations. There was no significant difference across groups on environmental attitude, F(2,155) = .006, p = .995, partial n2 = .000; or on environmental identity F(2, 155) = .433, p = .649, partial n2 = .006.”
Point 3: I appreciate that the authors transparently report the comparisons between their experimental conditions and the control condition on the manipulation check. But, the fact that there were no differences between experimental conditions and the control condition on the manipulation check puts into question the interpretation of findings. That is, if people in the cooperative priming condition did not score higher in cooperative values than those in the control condition, what does that tell us about any effects of the priming manipulation?
Response #3: This is the major issue of the study. Indeed, priming manipulation in the present study may not enhance the cooperative score since participants in the cooperative priming condition did not score significantly higher in cooperative values than those in the control condition. The findings of the study should be interpreted with caution.
We stated this issue as a major limitation on page 14.
“One major issue is that the priming manipulation in the present study may not enhance the cooperative score. It was found that people in the cooperative priming condition did not score higher in cooperative values than those in the control condition. The findings of the study should be interpreted with caution. It may be that the manipulation check by Lu et al. [31] assessed cooperation as a personality trait. In the current study, we considered cooperation as a state that can be primed. However, we were unable to test this assumption.”
Lu, S., Au, W.-T., Jiang, F., Xie, X., & Yam, P. Cooperativeness and competitiveness as two distinct constructs: Validating the Cooperative and Competitive Personality Scale in a social dilemma context. International Journal of Psychology, 2013, 48, 1135–1147.
Point 4: Finally, I think that the authors' argument about cooperative priming influencing participatory, but not leadership, actions makes sense, post-hoc. What would strengthen confidence in this finding would be to replicate their study, ideally pre-registering the hypotheses that cooperation versus competitiveness will specifically influence low-cost environmental actions rather than leadership actions.
Response #4:Thank you for mentioning this idea. We added a brief discussion on pre-registering the hypothesis for future researchers on page 16.
“Ideally, researchers can pre-register hypotheses stating that cooperation versus competitiveness will specifically influence low-cost environmental participatory actions rather than leadership actions.”
In sum, many thanks to the reviewer’s helpful comments. I hope we addressed all the concerns in this revision.

Reviewer 3 Report
The authors have included all the suggestions and have made all the changes that were recommended.
Author Response
Thank you very much.
Round 2
Reviewer 1 Report
No further comments.
Reviewer 2 Report
I thank the authors for considering my suggestions and for their responses.
This manuscript is a resubmission of an earlier submission. The following is a list of the peer review reports and author responses from that submission.
Round 1
Reviewer 1 Report
The manuscripts presents an interesting and emerging field of study with the adoption of an innovative game experiment; Is "game theory" part of the theoretical construct or is it a research methodology in the study? While I would assume the independent variables of the study are the social values (an individual's preference for cooperation or competition) and the dependent variables are the environmental actions (expressed in behaviour and leadership), I am unsure how the authors could adequately identify the participants representing their inclined social value as the game is designed in such a way that the 155 participants were randomly placed into one of the three conditions (cooperative, competitive or neutrally); Further to the above point (3), the hypothesis "it was expected that if participants were primed by writing cooperative stories, they would score higher in environmental action" was not being tested in the study. There seems to have little construct on the relationship between environmental leadership and social value.Reviewer 2 Report
This paper examines how priming individuals with cooperation versus competition influences intentions to act pro-environmentally. The authors report one study with a student sample (N = 155), in which participants were (1) first allocated to one of two experimental groups (i.e., cooperative priming; competitive priming) or a control group (2) then wrote about a cooperative, competitive, or neutral event, respectively, and (3) filled out a self-report measure of (future) pro-environmental action. Results suggest that the priming manipulation had no overall effect on intentions to act pro-environmentally. When intentions to engage in participatory actions and organizational leadership actions were analyzed separately, participants in the cooperative-prime condition reported higher pro-environmental intentions than participants in the competitive-prime condition, but not compared to those in the control group. The study deals with an important and pressing topic, but I think that, as it stands, the paper has important limitations, on which I elaborate below.
Introduction
Throughout the introduction, the authors refer to cooperation and competition quite broadly, without making it clear that they will be focusing on the effects of priming these constructs. Thus, at least for me, it took a while to understand what they were planning to do, based on rather broad statements, such as “the present study examined the relationship between a social dilemma (competition vs. cooperation) and environmental actions.” or “Promoting cooperation could be effective at increasing personal restraint in environmental decisions.” I think the whole paper would benefit from consistently and explicitly using the term ‘priming’ (cooperation vs. competition). The introduction could also be re-structured to avoid redundancy. For example, the definition of social dilemmas is repeated multiple times throughout, along with examples of social dilemmas/environmental problems, as is the definition of social value orientations (a construct which does not seem immediately relevant to the present study).Methods
In page 9, the authors state that “After the priming procedure, subjects took a battery of surveys to assess three dependent variables central to environmentalism. They were assessed upon environmental action, their environmental identity, and their environmental attitudes.” However, if I’m not mistaken, they only report findings with regards to effects of priming on environmental action. What about the other constructs/measures? It would be helpful if the authors gave more details about all the measures they included in the study and reasons to include/exclude measures from reported analyses. Manipulation check: The authors report that participants in the cooperative-prime condition reported higher cooperation than those in the competitive-prime condition. It would be helpful to also provide information about the mean and SD of the control group (and how this compared to experimental groups) to see if the manipulation was indeed successful. The study relies on self-reports of intentions to engage in pro-environmental behavior in the future. Unfortunately, as the authors note in the discussion, there can be strong demand and self-presentation effects when asking participants to report on their intentions to behave pro-environmentally. Additionally, previous work suggests that reported environmental concern is not strongly associated with actual pro-environmental behaviors (Steg & Vlek, 2009). I think one key step to examine the effectiveness of priming cooperation versus competition would be to investigate whether they influence actual pro-environmental behaviors (e.g., donations to environmental causes, offering to volunteer time for an environmental group, or even just providing contact details to be kept informed about environmental issues).Analyses
Finally, I have doubts about whether the results of the study indeed support the conclusions drawn. The authors find no overall effect of the priming manipulation on environmental action. Given the lack of a main effect, I would caution against interpreting post-hoc comparisons between groups. Then, when the authors analyze the environmental action sub-scales separately (was this planned a priori or was it purely exploratory?), they find a difference in intended pro-environmental action between the cooperative-prime group and competitive-prime group, but no difference between either of the experimental groups and the control group. Together, based on these findings, it seems safer to conclude that priming cooperation does not influence intentions to act pro-environmentally, at least compared to a neutral prime.Reference
Steg, L., & Vlek, C. (2009). Encouraging pro-environmental behavior: An integrative review and research agenda. Journal of Environmental Psychology, 29, 309–317.
Reviewer 3 Report
REPORT ON COOPERATION AND COMPETITION IMPACT ENVIRONMENTAL ACTION: AN EXPERIMENTAL STUDY IN SOCIAL DILEMMA
COMMENT 1.
Authors must follow formal norms of the journal.
COMMENT 2. The sections are not numbered. The text starts without a heading.
COMMENT 3. Lines 36-59. It can be understood that these lines cover the introduction. The introduction must be drastically improved. In a scientific paper, there can’t be general statements. They should rely on data and references. In addition, the references in the introduction should be updated, especially taking into account the issue at hand. COMMENT 4 Lines 61-103. Section. Social Dilemmas: Cooperation and Competition The review of the literature in this section must be updated and expanded. Moreover, the authors should mention the approach of social preferences applied to the tragedy of the commons. They should include the contribution of Monroy et al. (2017). In this paper, the agents act independently, and then the authors do not consider the possibility of institutional arrangements or collective actions for managing the common resource as proposed in Ostrom (1990). This paper achieves a strong conclusion: only a pro‐social agent is required to avoid the tragedy of the commons, and it is not conditional on other agents' pro‐social behavior, which differs from the conclusion of Ostrom (1998). This relevant result should be commented in the paper. Monroy, L.; Caraballo, M. A.; Mármol, A. Mª; Zapata, A. (2017). Agents with other-regarding preferences in the commons. METROECONOMICA. 68:947–965.References of Ostrom: in the above paper. COMMENT 5 Lines 104-123. Section. Environmental Action. In this Section, the authors should make a brief comment on empirical data about environmental action. They can use the available values survey
Afrobarometer.org. (2019). Afrobarometer. [online] Available at: https://www.afrobarometer.org/
Arabbarometer.org. (2019). Arab Barometer. [online] Available at: https://www.arabbarometer.org/
台大胡佛東亞民主研究中心 Asian Barometer. (2019). Home. [online] Available at: http://www.asianbarometer.org/
Eurobarómetro. (2019). Eurobarómetro. [online] Available at: https://www.europarl.europa.eu/at-your-service/es/be-heard/eurobarometer
Europeansocialsurvey.org. (2019). La ESS | European Social Survey (ESS). [online] Available at: https://www.europeansocialsurvey.org/about/country/spain/
European Values Study. (2019). European Values Study. [online] Available at: https://europeanvaluesstudy.eu/
Latinobarometro.org. (2019). Latinobarómetro Database. [online] Available at: http://www.latinobarometro.org/lat.jsp
Worldvaluessurvey.org. (2019). WVS Database. [online] Available at: http://www.worldvaluessurvey.org/wvs.jsp
COMMENT 6
In order to explain much better the concepts (line 105: Environmental action is often used interchangeably with the terms pro-environmental behaviors and environmental activism), they can use the following references that will help them to clarify the concepts:
Berenguer et al, J. (2002). Measuring environmental attitudes: Proposal for an environmental consciousness scaleEcobarometer. Intervención Psicosocial, 11, nº 3, pp.349-358.
Dunlap, R. and York, R. (2008). The Globalization of Environmental Concern and The Limits of The Postmaterialist Values Explanation: Evidence from Four Multinational Surveys. The Sociological Quarterly, 49(3), pp.529-563.
Franzen, A., & Meyer, R. (2009). Environmental Attitudes in Cross-National Perspective: A Multilevel Analysis of the ISSP 1993 and 2000. European Sociological Review, 26(2), 219–234.
Grubert, E. (2018). Relational values in environmental assessment: the social context of environmental impact. Current Opinion in Environmental Sustainability, 35, pp.100-107.
Jiménez Sánchez, M. and Lafuente, R. (2010). Defining and measuring environmental consciousness. Revista Internacional de Sociología, 68(3), pp.731-755.
Knight, K. W., & Messer, B. L. (2012). Environmental Concern in Cross-National Perspective: The Effects of Affluence, Environmental Degradation, and World Society*. Social Science Quarterly.
Litina, A., Moriconi, S. and Zanaj, S. (2016). The Cultural Transmission of Environmental Values: A Comparative Approach. World Development, 84, pp.131-148.
Milfont, T. and Duckitt, J. (2010). The environmental attitudes inventory: A valid and reliable measure to assess the structure of environmental attitudes. Journal of Environmental Psychology, 30(1), pp.80-94
Riley E. Dunlap & Richard York (2008) The Globalization of Environmental Concern and The Limits of The Postmaterialist Values Explanation: Evidence from Four Multinational Surveys, The Sociological Quarterly, 49:3, 529-563.
The above references can be also useful for the Section Relationship between Social Dilemma and Environmental Action
COMMENT 7
The main methodological concern is about sample representativeness. The sample size is 155 respondents. The authors should include a section explaining the methodology to select the sample and a reasoned justification to show that the sample is representative and therefore that the results make sense.